# Potential of the miR-200 Family as a Target for Developing Anti-Cancer Therapeutics

**DOI:** 10.3390/ijms23115881

**Published:** 2022-05-24

**Authors:** Hyein Jo, Kyeonghee Shim, Dooil Jeoung

**Affiliations:** Department of Biochemistry, College of Natural Sciences, Kangwon National University, Chuncheon 24341, Korea; qnfdudn1212@gmail.com (H.J.); sim991127@kangwon.ac.kr (K.S.)

**Keywords:** anti-cancer drug resistance, cancer/testis antigen CAGE, clinical value, microRNA 200 family, molecular network, mechanism, microRNA mimics, PD-L1

## Abstract

MicroRNAs (miRNAs) are small non-coding RNAs (18–24 nucleotides) that play significant roles in cell proliferation, development, invasion, cancer development, cancer progression, and anti-cancer drug resistance. miRNAs target multiple genes and play diverse roles. miRNAs can bind to the 3′UTR of target genes and inhibit translation or promote the degradation of target genes. miR-200 family miRNAs mostly act as tumor suppressors and are commonly decreased in cancer. The miR-200 family has been reported as a valuable diagnostic and prognostic marker. This review discusses the clinical value of the miR-200 family, focusing on the role of the miR-200 family in the development of cancer and anti-cancer drug resistance. This review also provides an overview of the factors that regulate the expression of the miR-200 family, targets of miR-200 family miRNAs, and the mechanism of anti-cancer drug resistance regulated by the miR-200 family.

## 1. Biogenesis of MicroRNAs

MicroRNAs (miRNAs/miRs) are noncoding single-stranded RNAs of 18–24 nucleotides in length. They can modulate gene expression through post-transcriptional control and are involved in cancer cell proliferation [1,2], apoptosis [3,4], invasion [5,6], metastasis [7], and anti-cancer drug resistance [8].

miRNAs are transcribed as primary miRNAs, which are subsequently cleaved to precursor miRNAs (pre-miRNAs) and further processed into mature single-stranded ~22-nt miRNAs (Figure 1). The biogenesis of miRNAs requires RNase III enzymes DROSHA and DICER1, members of the Argonaute family (AGO1–4), and RNA polymerase II [9,10,11]. Mature miRNAs regulate gene expression by the cleavage of mRNA, translational repression, and the recruitment of epigenetic modifiers such as histone deacetylases (HDACs) and histone methyltransferases (HMT) [9,10,11] (Figure 1). The deletion of miRNA biogenesis proteins can result in embryonic lethality [12]. This suggests the role of miRNA biogenesis in developmental processes. The downregulation of DICER1, a regulator of miRNA biogenesis, decreased the expression of E-cadherin and enhanced the invasion of endometrial cancer cells [13]. This implies the role of DICER1 in epithelial-mesenchymal transition (EMT). DICER1 was highly expressed in cisplatin-resistant non-small cell lung cancer cells and induced cisplatin resistance by promoting autophagy [14] (Figure 1). DICER1 promoted colon cancer cell invasion by increasing the small non-coding RNA tRF-20-MEJB5Y13 [15] (Figure 1). DICER1 targeted the miR-200 family to promote the development of colon cancer and metastasis [16] (Figure 1). The overexpression of DICER1 enhanced the invasion and migration of lung cancer cells [17] (Figure 1). These findings suggest that the biogenesis of miRNAs plays a critical role in cancer development, cancer progression, and anti-cancer drug resistance.

## 2. miR-200 Family as a Diagnostic and Prognostic Marker

The decreased expression of the miR-200 family has been reported in various cancers. miR-200b was reported to be frequently downregulated in renal cell carcinoma [18]. Low levels of miR-200 were associated with high-grade glioblastomas [19]. High expression levels of miR-200b and miR-200c were associated with the high overall survival (OS) and progression-free survival (PFS) of patients with clear cell renal cell carcinoma (ccRCC) [20]. Low levels of miR-200b predicted the poor survival of patients with gastric cancer [21]. The low expression of miR-200b was strongly associated with the poor survival of patients with hepatocellular carcinomas [22]. High levels of the miR-200 family were strongly associated with better survival of bladder cancer patients [23]. The high expression of miR-200b was correlated with better responses of lung cancer patients to immunotherapy involving anti-PD-l antibodies such as pembrolizumab [24].

Circulating miRNAs (miRs) have been increasingly recognized as potential biomarkers in cancer [25,26]. Serum levels of miR-141 and miR-200a were decreased in hepatocellular carcinoma and could predict cancer metastasis [27]. Serum levels of miR-200b were lower in patients with non-small cell lung cancers than in healthy controls [28]. Circulating miR-200b is higher in metastatic breast cancer than in early breast cancer [29]. High levels of circulating miR-200c and miR-141 were associated with the poor OS of colon cancer patients [30]. Circulating miR-200 family members could predict OS and PFS in patients undergoing systemic therapy for metastatic breast cancer [31]. High levels of miR200 family members were strongly associated with reduced OS and PFS [31]. These reports indicate that the miR-200 family could serve as both diagnostic and prognostic markers. The above reports also suggest that the miR-200 family could be a target for developing anti-cancer drugs. Table 1 summarizes the clinical value of the miR-200 family as diagnostic and prognostic markers.

## 3. Regulation of miR-200 Family Expression

miR-200 family miRNAs are among the most extensively studied miRNAs. The miR-200 family consists of miR-200a, miR-200b, miR-200c, miR-141, and miR-429. These five miRNAs are clustered in two chromosomal locations. miR-200a, miR-200b, and miR-429 are located on chromosome 1, and miR-141 and miR-200c are located on chromosome 12 (Figure 2A). Figure 2B shows the seed sequences of miR-200 family miRNAs. The seed sequences of miR-200b/-c/-429 show one nucleotide difference from the seed sequence of miR-141/-200a (Figure 2B).

The promoter sequences of miR-200a/-b/-c contain a potential binding site for p53 (Figure 3A). p53 is known to increase the expression of the miR-200 family [32]. The downregulation of miR-200c by p53 mutation induced the resistance of breast cancer cells to doxorubicin [33]. p53-binding protein 1 (53BP1) inhibited epithelial-mesenchymal transition (EMT) in breast cancer cells by directly regulating the expression levels of miR-200b and miR-429 [34]. P53 might directly bind to the promoter sequences of the miR-200 family.

The promoter sequences of miR-200a/-b/-c contain a potential binding site for paired box-5 (Pax-5) (Figure 3A). Pax-5 could inhibit the invasion and proliferation of breast cancer cells by suppressing EMT [35]. Pax-5 inhibited the proliferation of breast cancer cells by increasing the expression of miR-215 [36]. Since miR-200 family miRNAs mostly function as tumor suppressors, Pax-5 might increase the expression of the miR-200 family.

The promoter sequences of miR-200a/-b/-c contain a potential binding site for nuclear receptors such as glucocorticoid receptor alpha (GR-α) (Figure 3A). GR activation suppressed pancreatic tumor growth [37]. It might increase the expression of the miR-200 family. Estrogen receptor (ER)-positive breast cancer cells were shown to express higher levels of miR-200c than triple-negative breast cancer cells [38]. The destabilization of the progesterone receptor by insulin-like growth factor-II mRNA-binding proteins 2 and 3 (IMP2 and IMP3) decreased the expression of miR-200a [39]. The decreased expression of miR-200a was seen in triple-negative breast cancer cells [39]. The overexpression of peroxisome proliferator-activated receptor alpha (PPARα) increased the expression of miR-200c in hepatocellular carcinoma cells [40]. Therefore, it is necessary to examine whether PPARα can suppress cancer growth.

The promoter sequences of miR-200a/-b/-c contain a potential binding site for Yin Yang 1 (YY1) (Figure 3A). The expression of YY1 was inversely correlated with miR-200a in Burkitt′s lymphoma (BL) tissue [41]. The promoter sequences of miR-200a/-b/-c contain a potential binding site for CAAT/enhancer-binding protein (C/EBP)-β (Figure 3A). The downregulation of C/EBP-β induced cisplatin resistance in malignant pleural mesothelioma cells [42]. Thus, it is necessary to examine the binding of YY1 and/or C/EBP-β to the promoter sequences of miR-200a/-b/-c. YY1 and C/EBP-β might regulate the expression of the miR-200 family.

Long non-coding RNAs can regulate the development of cancer, cancer cell proliferation, and invasion/migration of cancer cells by binding to mRNA, miRNA, and protein [43]. Long non-coding RNA LncARSR bound to the miR-200 family and increased the expression of zinc finger E-box binding homeobox (ZEB1)/ZEB2, a master regulator of EMT to induce EMT in ovarian cancer cells [43]. Metastasis-associated lung adenocarcinoma transcript 1 (MALAT1) induced docetaxel resistance in breast cancer cells by decreasing the expression of miR-200b [44]. The expression of long non-coding RNA ZFAS1 was higher in colon cancer tissue than in normal tissue [45]. ZFAS1 targeted miR-200b/-c to promote EMT in colon cancer cells [45]. CCAT2, a long non-coding RNA, promoted the invasion and tumorigenic potential of esophageal squamous cell carcinoma (ESCC) by decreasing the expression of miR-200b [46]. Lnc-ATB, a long non-coding RNA, decreased the expression of miR-200c to promote cholangiocarcinoma (CCA) growth [47]. Lnc RNA H19 targeted miR-200a and promoted the invasion and migration of glioma cells [48].

Epigenetic modifications play critical roles in EMT and cancer cell proliferation [49]. The promoter regions of miR-200 clusters contain CpG islands that undergo DNA methylation [50,51,52,53]. Promoter methylation of miR-200b promoted the proliferation and invasion of endometrial cancer cells [54]. DNMT1 and enhancer of zeste homolog 2 (EZH2), a histone methyltransferase, could bind to the promoter sequences of miR-200b/a/429, leading to the downregulation of the miR-200 family [55]. Promoter methylation by MYC and DNMT3A decreased the expression of miR-200b in triple-negative breast cancer cells [56]. Therefore, it would be interesting to examine the effects of DNMT1 and DNMT3 A on EMT, cancer growth, and anti-cancer drug resistance. TargetScan analysis predicted histone deacetylase 4 (HDAC4) as a target of the miR-200 family. HDAC4 induced EMT and cancer stem cell-like properties in cancer cells [57]. The overexpression of HDAC4 decreased the expression of miR-200b, which led to the resistance of lung cancer cells to anti-cancer drugs [58].

MicroRNA-200b (miR-200b) is a downstream target of p38γ mitogen-activated protein kinase p38γ MAPK and is inhibited by p38γ MAPK [59]. The activation of Kindlin-2-integrin β1-AKT signaling was associated with the decreased expression of miR-200b in esophageal squamous cell carcinoma cells [60]. Hepatocyte nuclear factors (HNFs) were reported to bind to promoter sequences of miR-200b, increase the expression of miR-200b, and suppress the stemness of colorectal cancer cells [61]. HNF-1β promoted EMT and the tumorigenic potential of hepatocellular carcinoma cells by activating Notch signaling [62]. Figure 3B shows factors that regulate the expression of the miR-200 family.

## 4. miR-200 Family and Targets of the miR-200 Family

miR-200 family miRNAs mainly function as tumor suppressors [63,64]. Transgenic overexpression of the miR-200 family suppressed the development of mammary tumors [65].

miR-200b targeted CXCL12 and suppressed the invasion and metastatic potential of gastric cancer cells [66]. The overexpression of miR-200c repressed genes encoding immune suppressive factors, including CD274, HMOX-1, and GDF15 [38]. Blockade of CXCL12-CXCR4 signaling enhanced anti-tumor effects by inhibiting immune suppression in ovarian cancer [67].

miR-200b targets Neuregulin 1 (NRG1) to inhibit the invasion of gastric cancer cells [21]. miR-200b targeted Notch1 and inhibited the proliferation and tube-forming potential of human umbilical vein endothelial cells [68]. miR-200 deficiency activated Notch signaling and promoted the proliferation of cancer-associated fibroblasts and the metastatic potential of lung cancer cells [69]. TargetScan analysis predicted bone morphogenetic protein 4 (BMP4) as a target of the miR-200 family. BMP4 promoted the resistance of MDA-MB-231 cells to anti-cancer drugs by upregulating Notch signaling [70]. miR-200b-mimics inhibited p38γ MAPK-induced EMT [59]. miR-200b/-c targeted rho family GTPase 3 (RhoE) and inhibited the proliferation of non-small cell lung cancer cells [71].

miR-200b directly regulated the expression of high mobility group 3B (HMG3B) and inhibited the proliferation of hepatocellular carcinoma cells [22]. miR-200b and miR-200c suppressed the progression of glioblastoma by directly decreasing the expression of HMG3B [72]. E2F transcription factor 3 (E2F3) and ZEB1 are targets of miR-200b and regulated docetaxel resistance in lung adenocarcinoma cells [44].

EMT plays a critical role in the crosstalk of tumor cells within the microenvironment. EMT is a highly plastic program. In other words, the mesenchymal/EMT phenotype can revert to mesenchymal-epithelial transition. The miR-200 family can regulate EMT [73,74]. miR-200b plays a critical role in EMT by interacting with genes involved in EMT, including receptors, signaling, and the cell cycle [75]. The miR-200 family plays a major role in specifying the epithelial phenotype by preventing the expression of transcription repressors ZEB1 and ZEB2 (Smad-interacting protein 1) [76,77,78]. TargetScan analysis predicted ZEB2 as a target of the miR-200 family. miR-200b/200a/429 suppressed the metastatic potential of breast cancer cells by decreasing the expression of ZEB1 [76]. miR-200b/miR-429 decreased the expression of ZEB1/2 and inhibited the migration potential of oral squamous cell carcinoma (OSCC) cells [79]. miR-200b-3p and miR-429-5p suppressed the proliferation and invasion of triple-negative breast cancer cells by decreasing the expression of cyclinD1/CDK4/CDK6 [80]. The downregulation of cyclin D1 led to the inhibition of EMT and cell cycle arrest [81]. EMT-regulating transcription factors are known to regulate anti-cancer drug resistance [82]. The downregulation of SNAIL enhanced the sensitivity of prostate cancer cells to anti-cancer drugs [83]. EMT activated cancer stem cells (CSCs) resistant to chemotherapy and target therapy [84]. Figure 3C shows the targets of the miR-200 family and the roles of these targets in anti-cancer drug resistance.

## 5. Role of the miR-200 Family in Anti-Cancer Drug Resistance

miRNAs can regulate the responses to anti-cancer drugs [85]. miR-200a-3p was shown to target dual-specificity phosphatase 6 (DUSP6) and enhance the sensitivity of hepatocellular carcinoma cells to 5-fluorouracil (5-FU) [86]. miR-200c targeted tyrosine-protein kinase B (TrkB) to enhance the sensitivity of breast cancer cells to doxorubicin [87]. TrkB promoted the EMT of prostate cancer cells in an Akt-dependent manner [88]. miR-200b inhibited the tumorigenic potential and enhanced the sensitivity of lung cancer cells to cisplatin by inhibiting ERK/Akt signaling and targeting ribosomal protein S6 kinase 1 (p70S6K1) [89]. miR-429 enhanced the sensitivity of pancreatic cancer cells to gemcitabine by upregulating Akt-inhibited programmed cell death 4 (PDCD4) [90].

High levels of EMT markers (ZEB1/ZEB2) and low levels of the miR-200 family (miR-200a/-b/-c) resulted in the resistance of estrogen receptor (ER)-positive breast cancer cells to tamoxifen [91]. The downregulation of miR-200c restored EMT and conferred the resistance of prostate cancer cells to docetaxel [92]. miR-200b was the most significantly downregulated miRNA in doxorubicin-resistant breast cancer cells [93]. miR-200b targeted fibronectin 1 (FN1) and suppressed EMT phenotypes to overcome doxorubicin resistance in breast cancer cells [93].

DNMT1 directly decreased the expression levels of miR-200a/-b/-429 and inhibited the progression of gastric cancer and glioblastoma [55]. This indicates the role of epigenetic modifications and the miR-200 family in anti-cancer drug resistance. miR-200b and miR-200c synergistically enhanced the sensitivity of ovarian cancer cells to cisplatin by targeting DNA methyltransferase I (DNMT1) [94]. DNMT1 promoted the resistance of breast cancer cells to cisplatin by decreasing the expression of miR-200b [95]. Flap endonuclease (FEN1)/DNMT3a complex decreased the expression of miR-200a and promoted breast cancer cell proliferation [96]. This implies that miR-200a may enhance chemosensitivity in breast cancer cells.

The allelic loss of Beclin1, a marker of autophagy, has been reported in various cancers [97]. The high expression of autophagy markers was strongly associated with poor survival of hepatocellular cancer patients [98]. Autophagy induction promoted EMT in glioma cells [99]. These reports indicate the roles of autophagy in the development of cancers and anti-cancer drug resistance. Cancer cells may induce autophagy for survival in response to anti-cancer drugs [100,101]. Autophagy is positively associated with anti-cancer drug resistance [102,103]. The activation of phosphoinositide 3 (PI3K)/Akt signaling promoted autophagy and chemotherapy resistance in breast cancer cells [104]. Targeting autophagy is known to enhance the cytotoxic effects of trastuzumab on human epidermal growth factor receptor 2 (HER2)-positive gastric cancer cells [105]. Autophagy inhibition enhanced the sensitivity of non-small cell lung cancer cells to osimertinib by decreasing the expression of SRY-box transcription factor 2 (SOX2), a marker of stemness [106]. This implies that cancer stemness is closely associated with anti-cancer drug resistance.

ZEB1, a regulator of EMT, induced autophagy to cause anti-cancer drug resistance in breast cancer cells [101]. ZEB1 may induce chemotherapy resistance by promoting EMT. This also implies a negative regulatory role of miR-200s in autophagy and chemotherapy resistance. The expression of miR-200b was inversely correlated with autophagy-associated gene 12 (ATG12) in docetaxel-resistant lung adenocarcinoma cells [107]. The downregulation of ATG12 by miR-200b enhanced the chemosensitivity of lung adenocarcinoma cells to docetaxel [107]. miR-200b decreased the expression of ATG-5 and enhanced the sensitivity of breast cancer cells to cisplatin and docetaxel [108]. It is probable that the combination of a miRNA-mimic (miRNA-mimic) and an inhibitor of autophagy can overcome the resistance of cancer cells to anti-cancer drugs.

miR-200b inhibited the proliferation of skin cancer stem cells [109]. SOX2 is known to be a target of the miR-200 family in colorectal cancer [110]. miR-429 enhanced the sensitivity of gastric cancer cells to cisplatin by decreasing the expression of SOX2 and inhibiting PI3K/Akt/mammalian target of rapamycin (mTOR) signaling [111]. These reports further indicate the role of cancer stem cells in anti-cancer drug resistance.

Cancer-associated gene (CAGE), a cancer/testis antigen, could bind to SOX2 and regulate the cancer stem cell-like properties of breast cancer cells [108]. CAGE bound to HDAC2 and SNAIL and repressed the expression of p53 in anti-cancer drug-resistant melanoma cells [112] (Figure 4A). CAGE conferred resistance to epidermal growth factor receptor (EGFR)-TKIs (EGFR tyrosine kinase inhibitors) by promoting autophagy via binding to Beclin1 in non-small cell lung cancer cells [113]. Since p53 increases the expression of the miR-200 family [32], miR-200 family members might decrease the expression of CAGE. TargetScan analysis predicted miR-200b as a negative regulator of CAGE. Figure 4A shows the potential binding of the miR-200 family to the 3′ untranslated region (UTR) of CAGE. miR-200b and CAGE could form a negative feedback loop and regulate the response of melanoma cells to anti-cancer drugs [114] (Figure 4B). miR-200b exerted negative effects on cancer stemness by decreasing the expression levels of CAGE and autophagic flux in breast cancer cells [108]. miR-200b decreased the expression of EGFR and suppressed the migration of gastrointestinal stromal tumors [115]. CAGE bound to the EGFR and HER2 and regulated the responses of melanoma cells to trastuzumab and microtubule-targeting anti-cancer drugs [116] (Figure 4B). Therefore, the effects of other miR-200 family members on the responses of cancer cells to anti-cancer drugs targeting EGFR should be examined.

HDAC3 could bind to the promoter sequences of CAGE and inhibit the tumorigenic potential of anti-cancer drug-resistant melanoma cells [117] (Figure 4B). HDAC3 enhanced the sensitivity of anti-cancer drug-resistant melanoma cells to anti-cancer drugs by negatively regulating EGFR signaling and CAGE expression [117] (Figure 4B). miR-326 acted as a negative regulator of HDAC3 and enhanced the invasion and migration of melanoma cells [118] (Figure 4B). miR-326 and miR-200b formed a negative feedback loop to regulate the response of melanoma cells to anti-cancer drugs [118]. HDAC3 might regulate EMT and autophagic flux. It is probable that the miR-200 family and HDAC3 might form a positive feedback loop.

The downregulation of tubulin β3 by HDAC3 enhanced the sensitivity of melanoma cells to microtubule-targeting agents [119]. Tubulin β3 may confer resistance to anti-cancer drugs by promoting EMT and autophagic flux. The overexpression of miR-200c targets class III tubulin (TUBB3) restored the expression of E-cadherin and enhanced the sensitivity to microtubule targeting agents in ovarian cancer cells [120]. miR-200c also targeted ZEB1/ZEB2 and various mesenchymal genes (FN1 and QK1) [120]. Other members of the miR-200 family might regulate the expression of tubulin β3.

Immune checkpoint molecules such as programmed death ligand-1 (PD-L1) and PD-1 are known as targets for anti-cancer drug development [121]. Figure 4C shows the regulation of PD-L1 expression by the miR-200 family. miR-200a-3p and miR-200c-3p decreased the expression of PD-L1, suppressed the development of colorectal cancer, and promoted anti-cancer immune responses [121]. miR-429 directly targeted PD-L1 and suppressed gastric cancer cell proliferation [122]. PD-L1 served as a target of miR-200a in non-small cell lung cancer cells [123]. PD-L1 promoted EMT by preventing glycogen synthase kinase 3β (GSK3β) from degrading SNAIL in triple-negative breast cancer cells [124] (Figure 4C). PD-L1 bound to the EGFR and promoted TNF-related apoptosis-inducing ligand (TRAIL) resistance in gastric cancer cells [125] (Figure 4C). miR-429 targeted PD-L1 and enhanced the sensitivity of gastric cancer cells to TRAIL [125] (Figure 4C). CAGE inactivated GSK3β by binding to GSK3β. The inactivation of GSK3β led to the increased expression of cyclinD1, causing the resistance of melanoma cells to microtubule-targeting drugs [126] (Figure 4C). A CAGE-derived peptide (^269^GTGKT^273^) could bind to CAGE and prevent CAGE from inactivating GSK3β [126] (Figure 4C). The CAGE-derived peptide enhanced the sensitivity of melanoma cells to microtubule-targeting drugs [126] (Figure 4C). Additionally, the CAGE–derived peptide inhibited the binding of CAGE to the EGFR and enhanced the sensitivity of anti-cancer drug-resistant melanoma cells to gefitinib and trastuzumab [116]. These reports indicate the potential role of the miR-200 family-CAGE-EGFR-PD-L1 molecular network in the resistance to immune checkpoint inhibitors, EGFR-tyrosine kinase inhibitors (EGFR-TKIs), and other anti-cancer drugs.

TargetScan analysis predicted proline, glutamate, and leucine-rich *protein 1* (PELP1) as a target of the miR-200 family. PELP1 is overexpressed in various cancers. Its high expression contributes to the pathogenesis of triple-negative breast cancer [127]. The high expression of PELP1 was strongly associated with the poor survival of lung adenocarcinoma patients [128]. PELP1 downregulation by miR-200-mimic or miR-141-mimic decreased the metastatic potential of cancer cells [129]. PELP1 directly bound to the promoter sequences of miR-200a and recruited HDAC2, which decreased the expression of miR-200a in breast cancer cells [129]. The downregulation of PELP1 enhanced the efficacy of chemotherapy by suppressing signal transducer and activator of transcription 3 (STAT3)/vascular endothelial growth factor (VEGF) signaling in colorectal cancer cells [130]. The downregulation of PELP1 enhanced the sensitivity of breast cancer cells to genotoxic agents by suppressing the cell cycle and enhancing apoptosis [131]. Since CAGE could bind to HDAC2 in anti-cancer drug-resistant melanoma cells, it will be interesting to examine whether CAGE could bind to PELP1. PELP1 might regulate the expression of the miR-200 family and bind to CAGE to exert its effect on the responses of cancer cells to anti-cancer drugs.

The epigenetic silencing of miRNAs has been reported in various cancers [132,133] and played critical roles in anti-cancer drug resistance [134]. The role of DNMT1 in chemotherapy resistance has been reported [135,136]. The epigenetic silencing of the miR-200 family was strongly associated with acquired EGFR-TKI resistance in non-small cell lung cancer (NSCLC) cells [137].

EGFR signaling mediated autophagy and anti-cancer drug resistance in non-small cell lung cancer cells [138]. Osimertinib-resistant non-small cell lung cancer cells displayed activated ERBB2 [139]. The overexpression of miR-200c inhibited the proliferation of acquired EGFR-TKI-resistant non-small cell lung cancer cells [137] and overcame resistance to gefitinib by inhibiting PI3K/Akt signaling in non-small cell lung cancer cells [140]. The activation of the hedgehog (Hh) signaling mediated the resistance of non-small cell lung cancer cells to EGFR-TKIs (gefitinib, afatinib, and osimertinib) [141]. miR-200b enhanced the sensitivity of non-small cell lung cancer cells to erlotinib and cisplatin by inhibiting Hh signaling [142]. The combination of a miR-200-mimic and inhibitor of EGFR signaling might enhance the sensitivity of cancer cells to EGFR-TKIs. Table 2 summarizes the mechanism of anti-cancer drug resistance regulated by the miR-200 family.

## 6. Conclusions

The miR-200 family has been reported to play diverse roles in cell proliferation, cancer development, EMT regulation, autophagy, and anti-cancer drug resistance. Many studies have reported the decreased expression of the miR-200 family in cancer tissues [18,19], and anti-cancer drug-resistant cancer cells [93]. The expression levels of miR-200 family miRNAs were shown to predict responses to anti-cancer therapeutics [19,20,21,22,23,24]. Thus, miR-200 family miRNAs can be a target for anti-cancer drug development. The global identification of targets of miR-200 family miRNAs is necessary to better understand the role of miR-200 family miRNAs in the development of cancer and anti-cancer drug resistance. These miR-200 family miRNA targets can also be employed for developing anti-cancer drugs.

Since miR-200 family miRNAs are mostly decreased in anti-cancer drug-resistant cells compared to parental sensitive cells, it is necessary to develop therapeutic miR-200 family miRNA-mimics (miR-200-mimics) that can overcome the resistance to anti-cancer drugs. The overexpression of miR-200 family miRNAs has been reported to enhance chemosensitivity [107,108,111]. Drugs targeting miR-200 family-regulated genes can be combined with miR-200 family miRNA-mimics to overcome the resistance to anti-cancer drugs.

miR-200 family miRNAs have been shown to target PD-L1 [121,122,123]. The combination of miR-200 family miRNA-mimics and immune checkpoint inhibitors might enhance the therapeutic potential of immune checkpoint inhibitors. miR-200 family miRNAs could negatively regulate EGFR signaling [115]. Combination therapy involving miR-200 family miRNA-mimics and EGFR-TKIs could overcome the resistance to EGFR-TKI. miR-200b was shown to form a negative feedback loop with the cancer/testis antigen CAGE to regulate anti-cancer drug resistance [114]. A CAGE-derived peptide enhanced the sensitivity of melanoma cells to anti-cancer drugs by disrupting CAGE-GSK3β interaction [126]. As with CAGE, PD-L1 could bind to the EGFR and inactivate GSK3β [124]. Thus, combining a miR-200 family miRNA-mimic and a CAGE-derived peptide can overcome resistance to immune check inhibitors.

Since CAGE forms a negative feedback loop with miR-200b, it is necessary to identify small molecules that target CAGE. These molecules may enhance the therapeutic value of miR-200-mimics. Polyphenols increased the expression of miR-200b and inhibited the proliferation of skin cancer stem cells [109]. Small molecules that regulate the expression of miRNA have been reported in breast cancer cells by employing a graphene-based biosensor [143]. Chemicals that increase the expression of miR-200 family miRNAs might overcome the resistance to anti-cancer drugs.

miRNA-based therapeutics employing mimics or inhibitors can be a reasonable tool to treat cancer and overcome the resistance to anti-cancer drugs [144] and are under development. Most clinical trials involving miRNAs are in phase I or phase II trials. However, no clinical trials involving mimics of miR-200 family miRNAs are underway. Compared to small interfering RNAs (siRNAs), few miRNA-mimics are in clinical trials. Unlike siRNAs, miRNAs target multiple genes, making miRNAs attractive targets for developing anti-cancer therapeutics. However, miRNA-mimics may cause off-target effects. In addition, miRNA-mimics are unstable with difficulty penetrating cell membranes due to their negative charge. Thus, the modification of miRNA-mimics is necessary to improve their pharmacokinetics and pharmacodynamics. The efficient delivery of miRNA-mimics is critical for the successful development of miRNA-based anti-cancer drugs. Delivery systems include viral vectors, lipid nanoparticles, cationic lipids, cell-penetrating peptides, polymer-based vectors, and bacterial mini cell vehicles. In the case of miRNA-mimics, the mimic concentration is also critical for successful clinical trials.

As mentioned above, efforts have been made to improve the specificity, immunogenicity, and delivery of miRNA-mimics. Although miRNA-mimics have not been successful in clinical trials, various efforts to enhance the clinical value of miRNA-mimics will eventually overcome cancer and anti-cancer drug resistance.

## Figures and Tables

**Figure 1 ijms-23-05881-f001:**
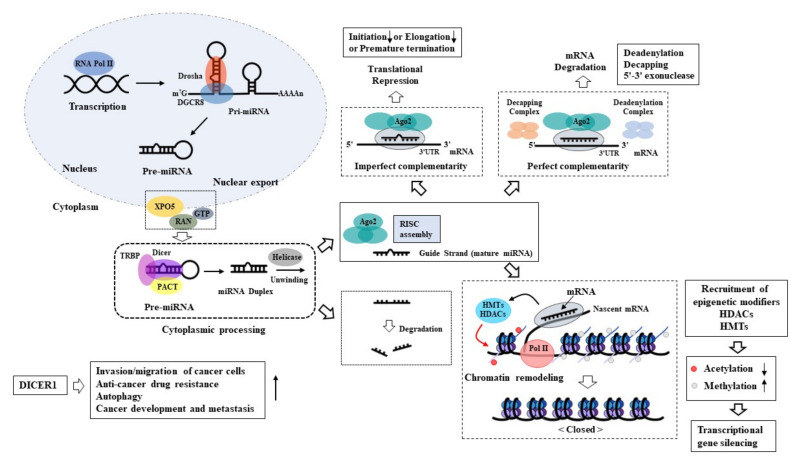
Biogenesis of miRNAs. Transcription to primary miRNA is catalyzed by RNA polymerase II or III. Primary miRNA is processed to precursor miRNA (~70 nucleotides) with a stem and loop structure by RNase Drosha and DiGeorge Critical Region 8 (DGCR8). Precursor miRNA is transported into the cytoplasm by the Exportin 5-Ran-GTP complex. Precursor miRNA is further processed to mature miRNA (~22 nucleotides) by RNase DICER. Double-stranded miRNA is unwound by helicase, and guide miRNA is incorporated into the RNA-induced silencing complex (RISC). Passenger miRNA is degraded. Perfect base-pairing between seed and target mRNA leads to the degradation of the target mRNA. The degradation of the target mRNA involves deadenylation, decapping, and 5′–3′ exonuclease activity. Imperfect base-pairing leads to translational inhibition. DICER1, a master regulator of miRNA biogenesis, promotes autophagy, anti-cancer drug resistance, autophagy, and cancer development. TRBP denotes TAR-RNA binding protein, HMT denotes histone methyltransferase, ↑ denotes positive regulation, ↓ denotes negative regulation. The arrows with solid lines indicate each step of miRNA biogenesis.

**Figure 2 ijms-23-05881-f002:**
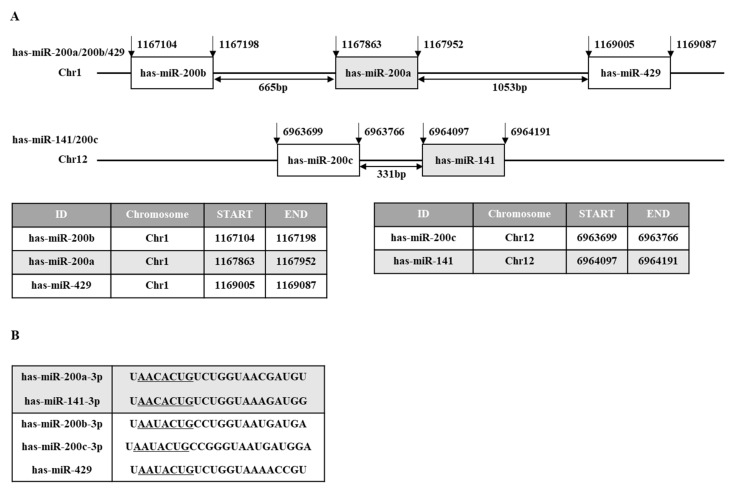
The chromosomal location and sequence of the miR-200 family. (**A**) The miR-200 family is divided into two clusters. Cluster 1 (human chromosome 1) contains miR-200a, miR-200b, and miR-429, and cluster 2 (human chromosome 10) contains miR-141and miR-200c. (**B**) The miR-200 family is divided into two groups based on the seed sequence. These two groups show a difference in the third nucleotide of the seed sequence. The seed sequences are underlined.

**Figure 3 ijms-23-05881-f003:**
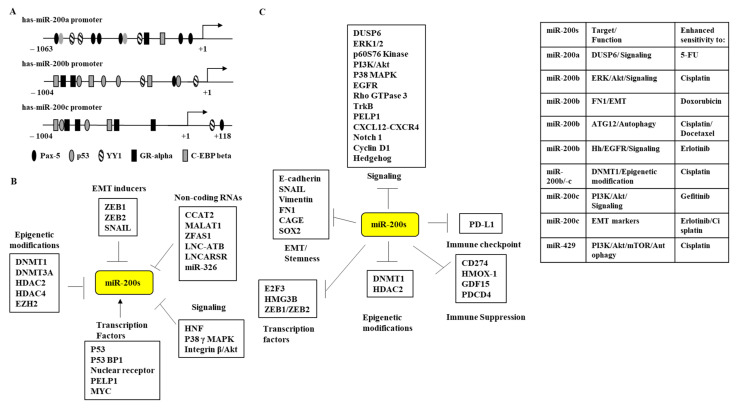
Regulation of miR-200 family expression and targets of the miR-200 family. (**A**) Promoter analysis revealed potential binding sites for various transcription factors in the promoter sequences of the miR-200 family. → denotes transcription activation. (**B**) Various factors regulating the expression of the miR-200 family. The T-bar arrows denote negative regulation, ↑ denotes positive regulation. (**C**) Targets of the miR-200 family and roles of these targets in anti-cancer drug resistance. The T-bar arrows denote negative regulation.

**Figure 4 ijms-23-05881-f004:**
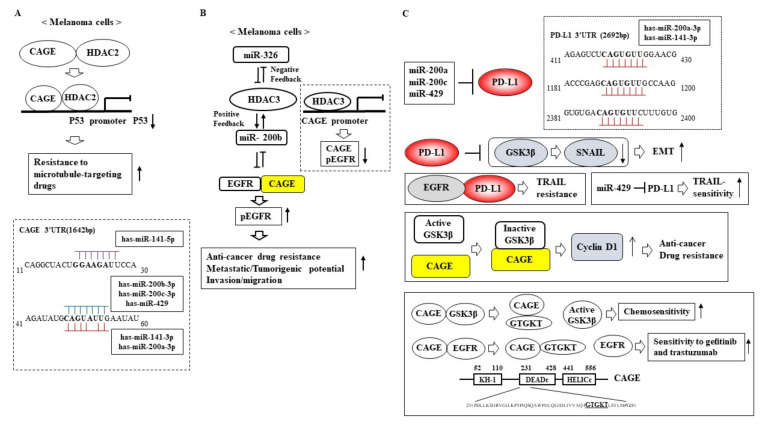
The proposed role of the HDAC3-CAGE-miR-200b-PD-L1 network in anti-cancer drug resistance. (**A**) CAGE binds to HDAC2 in anti-cancer drug-resistant melanoma cells. CAGE-HDAC2 complex binds to the promoter sequences of p53 to repress p53 expression. The decreased expression of p53 leads to resistance to anti-cancer drugs. ↓ denotes decreased expression. ↑ denotes positive regulation. The T-bar arrows denote transcription inhibition. The potential binding of the miR-200 family to the 3′ UTR of CAGE is shown. (**B**) HDAC3 forms a positive feedback loop with miR-200b and a negative feedback loop with miR-326. HDAC3 binds to the promoter sequences of CAGE, which decreases the expression of CAGE and pEGFR^Y845^. miR-200b forms a negative feedback loop with CAGE. The binding of CAGE to the EGFR leads to anti-cancer drug resistance and the enhanced metastatic potential of cancer cells. ↓ denotes decreased expression. ↑ denotes positive regulation. Both side T-bars denote negative feedback. (**C**) The miR-200 family decreases the expression of PD-L1. The binding of the miR-200 family to the 3′ UTR of PD-L1 is shown. PD-L1 induces EMT by preventing GSK3β from degrading SNAIL. miR-429 decreases the expression of PD-L1 to enhance sensitivity to TRAIL. CAGE binds to GSK3β to inactivate GSK3β. The inactivation of GSK3β leads to increased cyclin D1 expression and promotes anti-cancer drug resistance. CAGE-derived peptide (^269^GTGKT^273^) inhibits the binding of CAGE to GSK3β to enhance chemosensitivity. GTGKT also inhibits the binding of CAGE to EGFR. ↑ denotes positive regulation. The T-bar arrows denote negative regulation.

**Table 1 ijms-23-05881-t001:** Potential value of miR-200 family miRNAs as diagnostic and prognostic markers. ↑ denotes increased expression and ↓ denotes decreased expression.

Tumor Type	Sample Type/Size	Expression of miR-200s	Application	Reference
Renal cell carcinoma	Eighty tumor tissues and adjacent tissues	↓ miR-200b in tumor tissues	Prediction of metastasis, prognosis	[18]
Glioma	Eighty-nine glioma tissues and 41 non-tumor tissues	↓ miR-200b in tumor tissues	Diagnosis and prognosis	[19]
Clear cell renal cell carcinoma	Normal tissues (*n* = 23), primary tumor tissues (*n* = 194), metastatic tumor tissues (*n*= 10)	↓ miR-141, ↓ miR-200b in metastatic tumor tissues High levels of miR-200b and miR-200c → longer progression-free survival	Diagnosis and prognosis	[20]
Gastric cancer	Sixty tumor tissues and normal control tissues	High level of mR-200b → high overall survival	Diagnosis and prognosis	[21]
Hepatocellular carcinoma	371 tumor tissues	↓ miR-200b ↑ HMG3B in tumor tissues —miR-200b targets HMG3B	Diagnosis	[22]
Bladder cancer	Tumor tissues from 1150 patients	High level of the miR-200 family → better prognosis	Prognosis	[23]
Non-small cell lung cancer (NSCLC)	Tumor tissues from 60 NSCLC patients	High level of miR-200b → high progression-free survival	Prognosis in response to systemic immune therapy	[24]
Hepatocellular carcinoma	Whole blood from thirty patients and normal controls	↓ miR-141 ↓ miR-200a in serum of cancer patients	Diagnosis	[27]
NSCLC	Fifty cancer patients and 30 normal controls	↓ miR-200b in serum of cancer patients	Diagnosis	[28]
Breast cancer	Whole blood from early (137) and metastatic patients (110)	High levels of miR-200b and miR-200c in metastatic patients compared to early patients High level of miR-200b → shorter overall survival	Diagnosis and prognosis	[29]
Colon cancer	Plasma and exosomes from fifty resected patients	Low levels of miR-141 and miR-200c → longer overall survival	Diagnosis	[30]
Breast cancer	Serum (*n* = 47) from metastatic cancer patients	High levels of the miR-200 family (miR-141, miR-200a, miR-200b, miR-429) → reduction in overall survival and progression-free survival	Prognosis	[31]

**Table 2 ijms-23-05881-t002:** Role of miR-200 family miRNAs in anti-cancer drug resistance and the mechanisms associated with it.

miR-200 Family	Mechanism	Target	Enhances Sensitivity to	Cancer	Reference
miR-200a	Signaling	DUSP6	5-FU	Hepatocellular carcinoma	[86]
miR-200b	mTOR pathway	P70S6K1	Cisplatin	Lung cancer	[89]
	FAK/Src signaling	FN1	Doxorubicin	Breast cancer	[93]
	Autophagy	ATG-12	Docetaxel	Lung cancer	[107]
	Autophagy	ATG-5	Cisplatin Docetaxel	Breast cancer	[108]
	Autophagy/cancer stemness	CAGE	Microtubule-targeting agents	Melanoma	[112]
	Signaling	Hh signaling	Erlotinib	Lung cancer	[142]
miR-200b/-200c	EMT	C-MYB	Tamoxifen	Breast cancer	[91]
	Epigenetic modification	DNMT1	Cisplatin	Ovarian cancer	[94]
miR-200c	Signaling	TrkB	Doxorubicin	Breast cancer	[87]
	EMT	E-cadherin, SNAIL	Docetaxel	Prostate cancer	[92]
	EMT	TUBB3, ZEB1/ZEB2	Microtubule-targeting agents	Ovarian, Breast cancer	[120]
	Signaling	PI3K/Akt	Gefitinib	Lung cancer	[137]
miR-200c/-141	EMT	E-cadherin, Vimentin	Oxaliplatin	Ovarian cancer	[95]
miR-429	Cell death	PDCD4	Gemcitabine	Pancreatic cancer	[90]
	Cancer stemness	SOX2	Cisplatin	Gastric cancer	[111]
	Immune suppression	PD-L1	TRAIL	Gastric cancer	[125]

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
