# Peer review of "Potential of the miR-200 Family as a Target for Developing Anti-Cancer Therapeutics"

_ijms, 2022, doi:10.3390/ijms23115881_

Round 1
Reviewer 1 Report
- Overall nicely written review article.
- Pictures and tables are helpful additions to the article.
Author Response
Thank you for generous comment. I send revised version of the manuscript. Other reviewers suggest English editing. In this revision, I asked professionals handle this manuscript. I hope that this revision is satisfactory.
Reviewer 2 Report
The manuscript describes an interesting approach in pharmacological research. The use of in silico techniques clearly leads to considerable savings of time, money and animal lives; in addition, results are validated by different cell biology approaches and show to be reliable. For these reasons the work deserves publication in IJMS.
Minor points:
English editing is required, the manuscript should be revised by a native English speaker. Some sentences are hardly understandable.
In fig 2B the name of Y axis should be added.
Fig. 5. Data concerning the effect of compounds on HBMEC should be shown in parallel with the effect on U87 cells: even though these data can be deducted from following figures, to see the comparison here could be of use to readers.
Author Response
Q1. English editing is required, the manuscript should be revised by a native English speaker. Some sentences are hardly understandable.
Ans. Thanks for good suggestion. In this revision, I asked professionals handle this manuscript. I hope that this revision is satisfactory.
Q2. In fig 2B the name of Y axis should be added.
Ans. In this manuscript, figure 2B shows seed sequences of miR-200 family. I do not know whether there should be a Y axis. I am sorry. I do not see any reason for making changes to figure 2B. Please let me know.
Q3. Fig. 5. Data concerning the effect of compounds on HBMEC should be shown in parallel with the effect on U87 cells: even though these data can be deducted from following figures, to see the comparison here could be of use to readers.
Ans. In this manuscript, there is no figure 5. In this manuscript, I did not mention compounds, HBMEC, or U87 cells in this manuscript. I am sorry. I do not have answer to your comment. Please let me know.
Reviewer 3 Report
The manuscript described miR-200 family as a target for developing anti-cancer therapeutics. The authors showed its diverse roles in cell proliferation, development of cancers, regulation of EMT, autophagy, and anti-cancer drug resistance. Thus, these findings will be useful for cancer therapy. Therefore, the manuscript is not too excellent to be published. In other words, the manuscript is so excellent that it should be published.
Comments
(1) What kind of cells produced and released miR-200 family miRNAs to the cancer cells?
(2) Was circulating miR-200 family in exosomes?
(3) It was true that miR-200 family suppressed cancer but its targets such as DUSP6, P70S6K1, or FN1 largely differed, depending on cancer type (Table 2.). Why?
That is all.
Author Response
(1) What kind of cells produced and released miR-200 family miRNAs to the cancer cells?
Ans. Tumor microenvironment consists of cancer cells and various immune cells including macrophages. Exosomal miRNAs derived from tumor associated macrophages (TAMs) affect cancer cell proliferation. Exosomal miR-7 from TAMs suppresses the metastasis of epithelial ovarian cancer cells by inhibiting epidermal growth factor receptor (EGFR)/Akt/extracellular regulated kinase (ERK) pathway (Hu Y et al., 2017). Exosomal miR-142-3p (miR-200 family) from macrophages decreases the expression levels of stathmin-1 and insulin-like growth factor-1 receptor to inhibit the proliferation of hepatocarcinoma cells (HCCs) (Aucher et al., 2013). It is well known that cancer associated fibroblasts (CAFs) interact with cancer cells and affect cancer cell proliferation. miR-200 family mediates functional interaction between CAFs and cancer cells (Xue B et al., 2021). Epigenetic modification and repression of miR-200b in CAFs enhances invasion potential of gastric cancer cells (Kurashige J et al., 2015). It is probable that immune cells (mast cells, B cells, and T cells) also release miR-200 family to regulate cancer cell proliferation.
(2) Was circulating miR-200 family in exosomes?
Ans. Yes. Circulating miR-200 family is present in serum/plasma and exosomes. Please take alook at table 1.
(3) It was true that miR-200 family suppressed cancer but its targets such as DUSP6, P70S6K1, or FN1 largely differed, depending on cancer type (Table 2.). Why?
Ans. It is known that miR-200 family functions as both tumor suppressor and oncogene depending on cancer cell types. Mostly, miR-200 family works as tumor suppressor. In this manuscript, I describe miR-200 family as a tumor suppressor. DUSP6 and P70S6K1 serve as targets of miR-200a in hepatocellular carcinoma and lung cancer, respectively. DUPS6 and P70S6K1 may not serve as targets of miR-200 family in other cancer types. FN1 serves as a target of miR-200b in breast cancer cells. FN1 may not serve as a target of miR-200 family in other cancer cell types. DUSP6 (Overexpression of DUPS6 promotes proliferation of cancer cells expressing HER2, Kanda Y et al., 2021; Inactivation of DUPS6 impairs growth in NRAS and BRAF mutant cells, Ito T et al., 2021), P70S6K1 (ribosomal S6K1 activation promotes HeLa cell proliferation, Jeon et al., 2022; ribosomal S6K1 activation promotes invasion and tumorigenic potential of esophagus carcinoma, Wang L et al., 2021), and FN1 (Wang X et al., 2022; gastric cancer progression) are mostly involved in cancer cell proliferation. It is therefore reasonable that these molecules play important roles in anti-cancer drug resistance. Cancers are very heterogeneous. Tumor microenvironment consists of cancer cells, immune cells, stromal cells, and molecules that mediates interactions between cancer cells and immune cells and stromal cells. Each cancer is faced with its own microenvironment. It is therefore reasonable that downstream targets (including as DUSP6, P70S6K1, or FN1) of miR-200 family might be different depending on cancer cell types.